# A Cr Anti-Sticking Layer for Improving Mold Release Quality in Electrochemical Replication of PVC Optical Molds

**DOI:** 10.3390/mi10100702

**Published:** 2019-10-15

**Authors:** Yue Li, Guang Yang, Qihui Wu, Jun Cheng, Zhichao Hu

**Affiliations:** College of Mechanical and Energy Engineering, Key Laboratory of Energy Cleaning Utilization, Development, Cleaning Combustion and Energy Utilization Research Center of Fujian Province, Jimei University, Xiamen 361021, China; 13695504755@sina.cn (Y.L.); qihui_wu@xmu.edu.cn (Q.W.); 200561000091@jmu.edu.cn (J.C.); 200561000090@jmu.edu.cn (Z.H.)

**Keywords:** optical thin films, magnetron sputtering, nanoscale antisticking layer, release strength, surface roughness

## Abstract

This paper addresses the issue of mold release quality in an electrochemical replication of an optical polyvinyl chloride (PVC) mold, which has microlens array or microprisms array on its surface. The main idea is to deposit a nanoscale Cr thin layer as an antisticking layer on the PVC mold surface, followed by Ag film deposition as the conductive layer using magnetron sputtering, and finally, a nickel layer is electrochemically deposited on the Ag surface. By doing so, the upripping of the nickel mold from the PVC mold becomes easier, resulting in better mold release quality. The experiment results showed that when the Cr antisticking layer was used, the release strength between the nickel mold and the PVC substrate reduced from 1.94 N/cm to 1.43 N/cm, the surface roughness of the PVC substrate after mold release reduced from 0.60 μm to 0.55 μm, the surface roughness of the nickel mold reduced from 0.63 μm to 0.49 μm, the retroreflection coefficient of the nickel mold increased from 1600 cd·lx^−1^·m^−2^ to 2100 cd·lx^−1^·m^−2^, and the surface energy of the PVC substrate reduced from 31.47 mN/m to 15.53 mN/m.

## 1. Introduction

Thin optical films have been developed rapidly for their applications in new energy technologies, new media, and information technologies [1]. The thin optical films are often made by hot stamping or ultraviolet (UV) curing methods, and the microlens array or microprisms array is imprinted on the thin film’s surface [2,3,4]. The thin optical films can effectively improve optical efficiency and light utilization ratio, and thus, have been used as flexible display film substrate and a solar cell substrate [5,6]. The length scale of a single microlens or a single microprism is in the range of a few tens of micrometers to a couple of hundred micrometers [7]. The shape accuracy, structural defects, and residual stress affect the geometric and physical properties of thin optical films [8,9].

The microlens array or microprisms array on the thin optical films is formed by imprinting the cavities of the nickel mold, which is obtained by electrodeposition of nickel on an original mold [10]. Therefore, the shape accuracy, dimensional accuracy, and surface finishing of the nickel mold are the source of optical thin film quality assurance [11,12,13,14].

One of the key technologies for replicating the mold is to control the mold release quality [15]. Lee et al. studied the process of preparing a thick, electroformed micromold by using a KMPR (commodity manufactured by Kayaku Microchem Co. Ltd., Tokyo, Japan) negative tone photoresist, and microelectroforming was performed on the KMPR mold. The results showed that the electroformed sample was not only good in geometry, but also easy to release, and did not damage the surface [16]. Zhang et al. studied the release properties of nickel and nickel-polytetrafluoroethylene composites. In the nickel plating solution, polytetrafluoroethylene was added for electroforming to obtain a perfect microstructure and the friction force during the release process was greatly reduced [17]. With respect to mold release quality, the consensus is that better mold release quality can be obtained with lower adhesion energy under the premise of ensuring normal electrodeposition. In order to reduce adhesion energy, the researchers put forward many methods, such as: EBeam evaporation [18], thermal spray coating [19], liquid phase deposition [20], superhydrophobic surface fabricated by machining [21], self-assembled molecular membrane method (SAM) [22], etc.

The anti-sticking layer must be thin and uniform, in order to ensure the optical properties [23,24,25]. The traditional coating is thick and unevenly distributed, for example, a Cu-coated mold with a thin layer of inorganic release agent, such as potassium dichromate, before electrodeposition of Ni [26]. The optical properties of the microlens array or microprisms array can be destroyed by machining the superhydrophobic surface [21]. Due to the fact that self-assembly (SAM) requires that the components be mobile, it usually takes place in fluid phases [27]. Therefore, SAM method is not suitable for the metal deposition on PVC. In addition, the thickness of PVC optical mold is only 0.2 mm, and the increase of temperature during the Cr film growth in EBeam evaporation and other thermal spraying methods will lead to its deformation, which seriously affects the optical properties of microprisms [28]. Magnetron sputtering has the characteristics of simple process and easy control. The deposition at near room temperature can be realized by magnetron sputtering. Therefore, magnetron sputtering method was selected to deposit the antisticking layer of PVC mold in the current study [29,30]. In this study, the 3 nm thickness of Cr antisticking layer was fabricated by magnetron sputtering. The release strength between the nickel mold and the original mold, the surface roughness, and reflective performance of the nickel mold with and without the Cr antisticking layer were studied and compared, and the explanations are provided.

## 2. Experiment Description 

### 2.1. Materials and Experiment Steps

The material of the original optical mold was PVC. Cr was selected as an antisticking layer material. Ag was selected as a cathode material, and Ni was selected as an anode material. The sputter targets was Cr and Ag with 99.999% purity, respectively. The purity of metal Ni anode for electrodeposition was 99.99%.

The antisticking layer was obtained by magnetron sputtering, which is schematically explained in Figure 1 describing the experimental process. First, the original PVC mold was selected and then was washed for 15 minutes with pure alcohol and pure water under ultrasonic agitation. As shown in Figure 1a, a thin layer of Cr with an average thickness of 3 nm was deposited on the surface of the original PVC mold by magnetron sputtering. Before sputtering, the chamber was pumped down to 2.5 × 10^−2^ Pa, and during sputtering, Ar gas was introduced into the vacuum with the vacuum going up to 1.0 Pa. After optimization of the experimental parameters, the sputtering power was 200 W [31,32] and the sputtering temperature was 30 °C. The sputtering time of the Cr antisticking layer was 15 s, and the sputtering time of the Ag conductive layer was 300 s. A layer of Ag with an average thickness of 500 nm was sequentially deposited on the Cr thin layer by magnetron sputtering, as well as the conductive layer, as shown in Figure 1b. Next, the Cr-Ag layers were used as cathode and a Ni plate was used as anode for electrodeposition in the nickel sulfonate plating solution, as shown in Figure 1c. The average thickness of electrodeposition layer was 0.3 mm. Finally, when the electrodeposition was completed, the nickel mold was demoulded, as shown in Figure 1d.

This work mainly studied the releasing effect under different schemes. The process parameters and thickness of deposition of Cr on PVC and Ag on PVC were different. The Cr layer was relatively thin, which can reduce the surface energy and help releasing. The Ag layer was thicker, and its role was to act as the conductive layer of electroforming Ni. In order to explain the effect of the Cr antisticking layer, two schemes were used and repeated three times. Scheme 1: Ag was sputtered as a conductive layer and then Ni was electrodeposited. Scheme 2: Cr was sputtered as an antisticking layer, then Ag was sputtered as a conductive layer, and, finally, Ni was electrodeposited.

### 2.2. Experiment Parameters

The sputtering parameters of Cr and Ag are shown in Table 1, and the electrodeposition parameters are shown in Table 2.

### 2.3. Experiment Characterizations

#### 2.3.1. Release Strength

As schematically illustrated in Figure 2, peel tests measure the force required to peel off the film per unit width, that is, it evaluates the release strength in terms of N/cm, which is equivalent to the energy required to peel off a unit area, J/m^2^. The nickel mold is gradually separated from the original PVC mold with the action of vertical force *F*. The average value of release force *F* was recorded by the dynamometer, and the release strength was calculated by σ = *F*/*L* [N/cm] [15,33].

#### 2.3.2. Surface Roughness

The surface roughness of the original PVC mold, the anti-sticking layer before electrodeposition, and the nickel mold after separation were measured by a VK-X100 laser confocal microscope (Keyence Corporation of America, Itasca, IL, USA). Three horizontal positions and vertical positions were selected for each sample, and the average values of the six roughness measurements were calculated as the surface roughness.

#### 2.3.3. Microscopic Morphology

The surface morphology was measured by a scanning electron microscope (Phenom-XL, Thermo Fisher Scientific, Eindhoven, The Netherlands) and a field emission scanning electron microscope (Sigma-300, Carl Zeiss, Oberkochen, Germany). Every sample was dried before measuring. Following this, 5 kV ultra-high pressure and backscattering were selected for measurement by a Phenom electron microscope. Then, 5 kV ultra-high pressure and in-lens mode were selected for measurement by a Sigma 300 field emission electron microscope. Three different positions of the test samples were photographed, and the pictures were saved with different multiples.

#### 2.3.4. Surface Energy

The surface energy was measured by a Theta Lite (Biolin Scientific, Gothenburg, Sweden), an optical contact angle measuring instrument. During the test, the sample was placed on the test platform, and the pure water (polar) and diiodomethane (non-polar) droplets with a volume of 5 μL were observed on the surface of the sample at room temperature. The testing liquid parameters are shown in Table 3. The automatic image method was used to measure the contact angle. The droplet profile, in accordance with the circular equation or elliptical equation, was fitted by the least square method, and then the contact angle of the droplet on the solid surface was calculated. For rough and microprism array surfaces, as well as considering incomplete infiltration, the Wenzel model and the Cassie–Baxter (CB) model were used to modify the contact angle. The surface energy was calculated by the Owens–Wendt–Rabel–Kaelble (WORK) model. Five different positions of every sample were measured to record the data, and finally, the average value was taken.

The contact angle formed by liquid phase on a solid surface was determined by the equilibrium of three interfacial tensions. The calculation of solid surface energy by contact angle was based on the Young’s equation [34,35];
(1)$$\gamma_{\mathrm{sg}}=\gamma_{\mathrm{sl}}+\gamma_{\mathrm{lg}}\cdot \cos\theta_{0}$$
where $\gamma_{\mathrm{sg}}$ means the interfacial tension of solid–gas, $\gamma_{\mathrm{sl}}$ means the interfacial tension of liquid–solid, $\gamma_{\mathrm{lg}}$ means the interfacial tension of liquid–gas, and $\theta_{0}$ is the contact angle. The formula (1) is suitable for smooth surfaces, and the actual sample needs to consider the effect of surface roughness. The effect of roughness correction factor *r* on contact angle was considered in this study. The $\theta_{0}$ was modified by using the Wenzel equation [36]:(2)$$\cos\theta_{w}=r\cdot \cos\theta_{0}.$$

If the rough factor $r_{f}$ of the wetting part is introduced, the CB equation in the metastable state of the droplet can also be obtained [37,38];
(3)$$\cos\theta_{\mathrm{CB}}=r_{f}\cdot f_{\mathrm{SL}}\cos\theta_{0}+f_{\mathrm{SL}}-1$$
where $f_{\mathrm{SL}}$ represents the percentage of contact area of solid–liquid, and $f_{\mathrm{SL}}$ = 0.95 and $r_{f}$ = 1.9 in Equation (3), were used in this study.

In terms of surface energy calculation, Owens and Wendt considered the effect of hydrogen bond, which is on the liquid–solid interface, and the following expression was obtained [39];
(4)$$\gamma_{\mathrm{sl}}=\gamma_{\mathrm{sg}}+\gamma_{\mathrm{lg}}-2\sqrt{\gamma_{\mathrm{sg}}^{d}\cdot \gamma_{\mathrm{lg}}^{d}}-2\sqrt{\gamma_{\mathrm{sg}}^{p}\cdot \gamma_{\mathrm{lg}}^{p}}$$
where, $\gamma_{\mathrm{sg}}^{p}$ and $\gamma_{\mathrm{sg}}^{d}$ are dispersion components and polar components of solid surface energy, and $\gamma_{\mathrm{lg}}^{d}$ and $\gamma_{\mathrm{lg}}^{p}$ are dispersion components and polar components of liquid surface energy, respectively. If formula (4) is combined with formula (1), called Owrk equation, one has;
(5)$$\sqrt{\gamma_{\mathrm{sg}}^{d}\cdot \gamma_{\mathrm{lg}}^{d}}+\sqrt{\gamma_{\mathrm{sg}}^{p}\cdot \gamma_{\mathrm{lg}}^{p}}=0.5\gamma_{\mathrm{lg}}\left(1+\cos\theta_{w}\right).$$

For Equation (5), there are two unknowns $\gamma_{\mathrm{sg}}^{d}$ and $\gamma_{\mathrm{sg}}^{p}$. The surface energy of the original mold was measured by water and diiodomethane, respectively.

#### 2.3.5. Retro-Reflection Coefficient

The retro-reflection coefficient of the nickel mold after the mold was released was measured by a STT-101 retro-reflection mark measuring instrument. Calibration conditions: incident angle is β = −4°, observation angle is α = 0.2°. The average value of five positions on a sample was obtained.

## 3. Results

### 3.1. Surface Energy of the Original Mold

The surface energy of the original mold with microprisms array was calculated by the contact angle. Considering the air retention in the V-shaped tank on the surface, the contact angles were corrected by the CB model. The calculated values are shown in Table 4. The average surface energy of the original PVC mold was 31.47 mN/m, and the average surface energy of the sputtered original mold was 15.53 mN/m. It shows that the surface energy of the original PVC mold can be reduced after sputtering an antisticking layer.

### 3.2. Roughness and Morphology under Two Different Sputtering Conditions

In this study, the surface roughness and the morphology of the original mold before electrodeposition under the two schemes were measured. Figure 3a and Figure 4a show the morphology of the original PVC mold, which had a surface roughness of *R*a = 0.60 μm. Figure 3b and Figure 4b show the morphology of PVC mold after deposition of the Cr films, whose surface roughness reached *R*a = 0.55 μm. Surface quality of original mold with the Cr antisticking layer was better than that of without the Cr anti-sticking layer, evidently. One may see that after deposition of Cr films, the sunken holes presenting on the PVC mold surface were partially filled up by the Cr films.

At the same time, the micromorphology of the Ag surface sputtered on the original mold under different deposition schemes was compared. The Ag surface micromorphology, detected with a field emission scanning electron microscope, is shown in Figure 5. Figure 5a,b correspond to the surface morphology of the original mold under scheme 1, the average size of crystal particles on the surface was about 220 nm. Figure 5c,d correspond to the surface morphology of the original mold under Scheme 2, and the average size of the crystal particles on the surface is about 150 nm.

In order to compare the surface micromorphology of the nickel mold after mold release under different deposition schemes, the micromorphology of the nickel mold after electrodeposition is shown in Figure 6. Figure 6a shows the Ni film’s micromorphology following scheme 1, and the surface roughness reached *R*a = 0.63 μm as presented in Figure 7a. Figure 6b displays the micromorphology of the Ni film according to scheme 2, and the surface roughness was *R*a = 0.49 μm as indicated in Figure 7b. It is obvious that the surface quality of the nickel mold with sputtered Cr antisticking layer was better than that without any Cr anti-sticking layer.

The surface micromorphology of the nickel mold after electrodeposition was observed by a field emission scanning electron microscope in Figure 8. The average size of crystal particles on the surface was about 230 nm under scheme 1, and about 100 nm under scheme 2. The crystal particle size has certain influence on the surface roughness of the nickel mold, and the roughness of the scheme 2 mold was smaller than that of scheme 1. The two surface components after mold release were measured by energy dispersive spectrometer (EDS) for scheme 2, and it was found that part of the Cr film was left on the side of the original mold and part was on the side of nickel mold, as shown in Figure 9a,b.

### 3.3. Release Strength

The average release force needed to separate the nickel mold from original PVC mold was measured and then the release strength was calculated. The average value of five measurements is shown in Table 5. According to the measurement results, the release strength between the nickel mold and the original PVC mold in scheme 1 was 1.94 N/cm. The release strength between the nickel mold and the original PVC mold in scheme 2 was 1.43 N/cm. The release strength of scheme 2 was obviously smaller than that of scheme 1.

### 3.4. Retro-Reflection Coefficient

One of the standards to measure the quality of a nickel mold is its reflective performance. In this paper, the retro-reflection coefficient of the nickel mold under two schemes was measured by a retro-reflection mark measuring instrument. In Table 6 the average retro-reflection coefficient measured by scheme 1 was 1600 cd·lx^−1^·m^−2^, and the retro-reflection coefficient under scheme 2 was 2100 cd·lx^−1^·m^−2^. It can be seen that the reflective performance of the nickel mold was improved by the Cr antisticking layer.

Combining the test results of the three times, we can see that the surface roughness of the original PVC mold was reduced by 8.33%, the release strength was decreased by 26.29% after electrodeposition, the surface roughness of the nickel mold was reduced by 22.22%, and the retro-reflection coefficient was increased by 31.25% with the sputtered Cr layer. The sum of surface energy was reduced by 10.99% with the sputtered Cr layer. The comparison results are shown in Figure 10.

## 4. Discussion

### 4.1. Mechanism of Anti-Sticking Layer Conducive to Demolding

The surface energy and the surface force derived from the surface energy determine the adhesion energy between the bonding surfaces. According to the fracture energy balance criterion, the intrinsic interface binding energy is equal to the sum of the actual interface binding energy and the residual strain energy [40]. The adhesion energy can be obtained by the sum of the two separated surface energies [41]. By comparison of surface energy of separated interfaces, as shown in Figure 10, the sum of surface energy between the substrate and the nickel mold was 62.33 mN/m in scheme 1 and the sum of surface energy between the substrate and the nickel mold was 56.16 mN/m in scheme 2. It can be seen that the adhesion energy of scheme 2 was smaller, which is more beneficial for releasing.

In addition, the reason why the thin Cr layer changes the surface energy can be understood from the micromorphology [42,43]. When Cr was deposited on the PVC material by magnetron sputtering, Cr particles formed island nucleation, as shown in Figure 11. Micro-nano hierarchical structure characteristics were formed with the original microprisms array of the thin optical films. Therefore, the surface energy of PVC optical thin films could be obviously reduced [44].

### 4.2. Effect of Anti-Sticking Layer on Surface Roughness of Original Mold and Nickel Mold

The bonding between the electrodeposition layer and the original mold mainly includes metal bond binding, intermolecular force, and mechanical bonding [45]. The factors that determine the interfacial bonding between an electrodeposition layer and an original mold include original mold type and original mold roughness. The true surface area of the electrodeposition layer in contact with the original mold is at atomic level, including the fluctuation and cracks at atomic level on the surface. Reducing the real contact surface area between the electrodeposition layer and the original mold can reduce the mechanical chimerism between the deposition layer and the original mold, and consequently, change the interfacial bonding properties [46,47].

By measuring the surface roughness of the PVC original mold and the thin Cr layer, it was found that the average roughness decreased from *R*a = 0.60 μm to *R*a = 0.55 μm, which indicates that the surface of the original PVC mold was smoothed by the Cr thin layer. Microscopic observation shows that because the Cr crystal particles were relatively small, some Cr filled the defects of the PVC film itself, as shown in Figure 5, which played a role in smoothing the surface of the original PVC mold.

After separation, the Cr thin layer was attached to one side of the nickel mold with the Ag conductive layer, and the surface morphology of the nickel mold after separation was observed. It showed that the smaller Cr particles were similar to the “inlay model” between the larger Ag particles, and the schematic illustration of the “inlay model” is shown in Figure 12 (the red thick line represents the profile of the surface of the Ni mold after the sputtering of Cr and Ag, and the blue thick line represents the surface outline of the direct sputter Ag on the nickel mold). This kind of mosaic plays a role in smoothing the surface of the nickel mold at the micro level.

In addition, if Ag is directly sputtered on the surface of PVC mold, it belongs to the physical deposition, and the position of Ag atoms or molecules is determined by sputter and will not change. However, with a layer of Cr, the position of Ag atoms or molecules can move microscopically due to the force between metals, and the Ag atoms are located at the lowest energy [48]. Such microlateral movement also results in a filling effect on the pits of the original mold.

## 5. Conclusions

In this study, a Cr antisticking layer was added using magnetron sputtering in the electrodeposition of nickel metal on a thin optical PVC film that had microlens or microprisms array on its surface. By comparing the release strength and surface roughness of the nickel mold with and without a Cr antisticking layer, the following conclusions were drawn:

(1) The nanothick Cr antisticking layer can reduce the release strength of the electrochemically deposited nickel mold from its PVC mold and improve surface roughness of the deposited nickel mold and the PVC mold as well. Hence their optical properties are improved.

(2) Micro-nano composite structure is one of the reasons for the reduction of adhesion energy of the nanometer antisticking layer, and the test results of the surface energy also support this conclusion. In addition, the surface roughness of the nickel mold was reduced by the inlay effect of the nanothickness Cr antisticking layer. Whether or not it has an inlay effect can be used as a condition for the selection of antisticking layer materials.

(3) Preparation of the nanothick Cr antisticking layer by magnetron sputtering is simple. It does not change the shape and dimensional accuracy of microlens array or microprisms array, and is suitable for optical film nickel mold. We believe that the separation process introduced in this study can be widely used in electroforming processes, which require ultraprecise accuracy.

## Figures and Tables

**Figure 1 micromachines-10-00702-f001:**
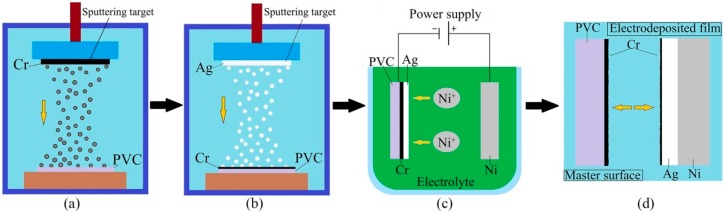
Experimental process. (**a**) Formation of release layer; (**b**) Formation of conductive Ag layer; (**c**) Formation of electrodeposition film; (**d**) Demoulding.

**Figure 2 micromachines-10-00702-f002:**
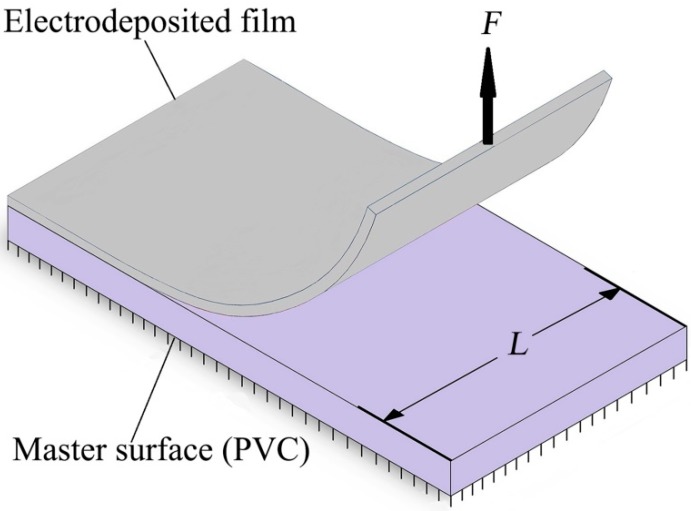
Schematic illustration of release test.

**Figure 3 micromachines-10-00702-f003:**
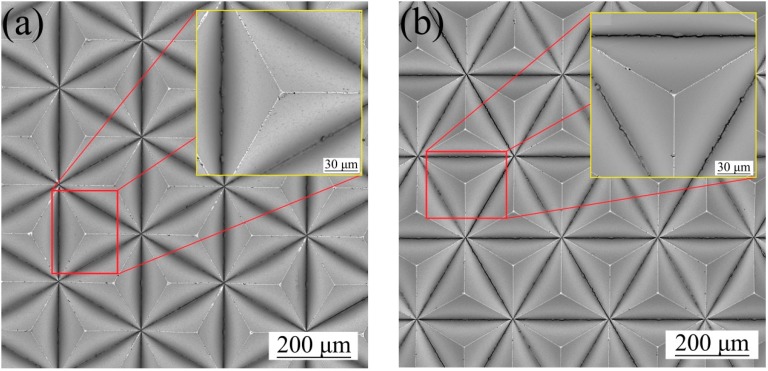
The surface morphology of original mold: (**a**) scheme 1; (**b**) scheme 2.

**Figure 4 micromachines-10-00702-f004:**
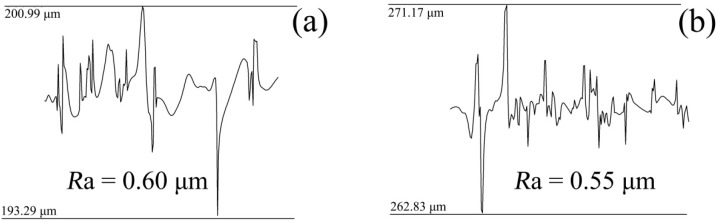
The roughness of original mold: (**a**) scheme 1; (**b**) scheme 2.

**Figure 5 micromachines-10-00702-f005:**
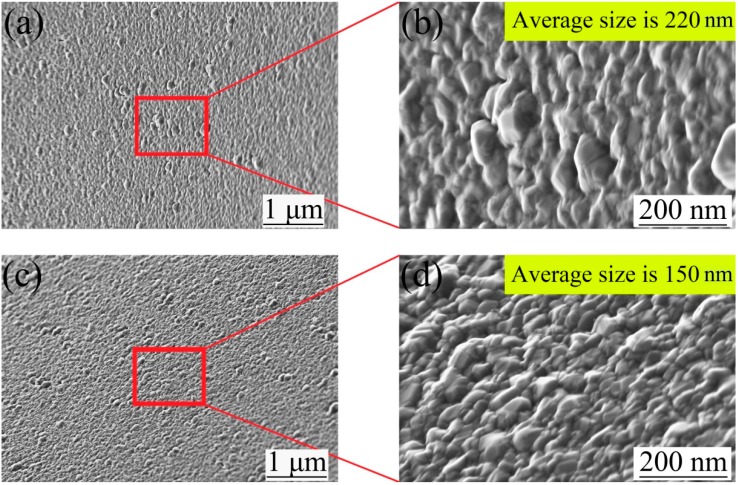
Ag surface micromorphology. (**a**,**b**) Scheme 1; (**c**,**d**) scheme 2.

**Figure 6 micromachines-10-00702-f006:**
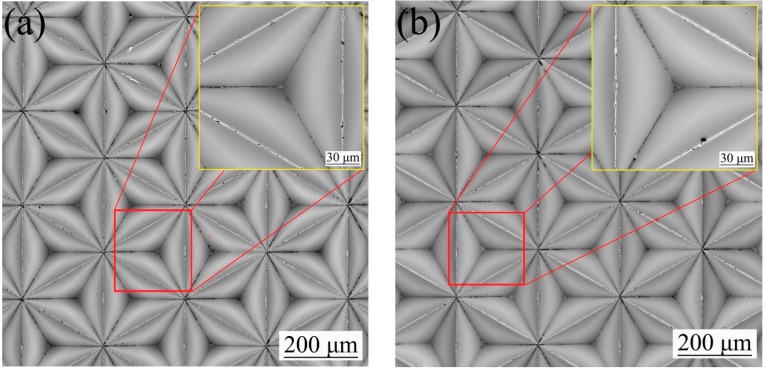
The micromorphology of the nickel mold: (**a**) scheme 1; (**b**) scheme 2.

**Figure 7 micromachines-10-00702-f007:**
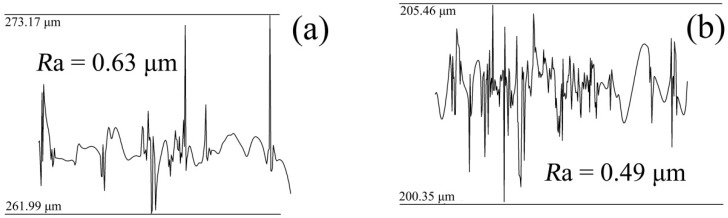
The roughness of the nickel mold. (**a**) Scheme 1; (**b**) scheme 2.

**Figure 8 micromachines-10-00702-f008:**
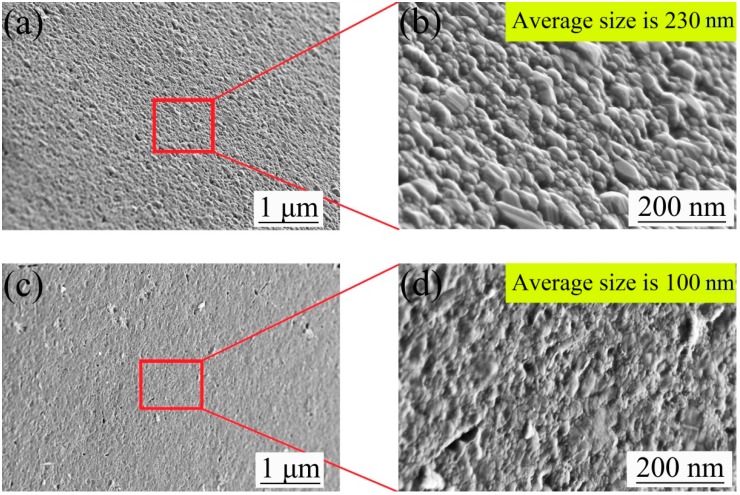
(**a**,**b**): The micromorphology of nickel mold after mold release under scheme 1; (**c**,**d**): The micromorphology of nickel mold after mold release under scheme 2.

**Figure 9 micromachines-10-00702-f009:**
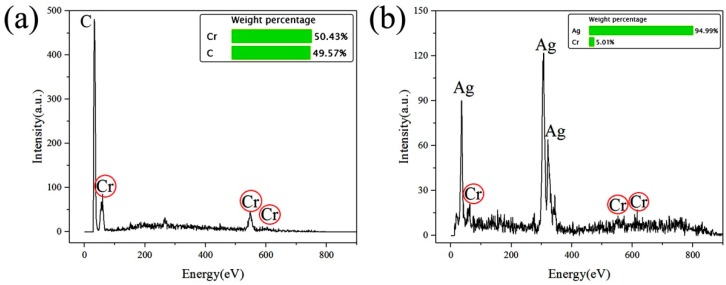
Energy dispersive spectrometer (EDS) images. (**a**) Surface components of original PVC mold side; (**b**) surface components of the nickel mold side.

**Figure 10 micromachines-10-00702-f010:**
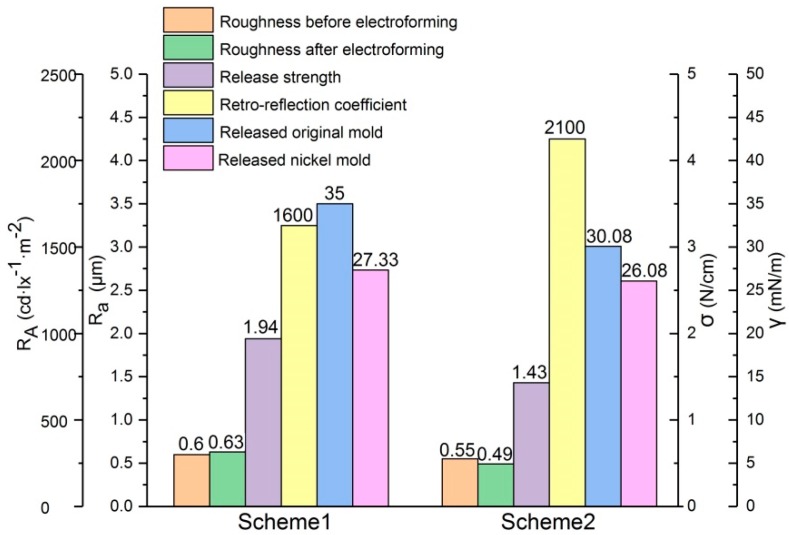
Comparison of the results of the two experiment schemes.

**Figure 11 micromachines-10-00702-f011:**
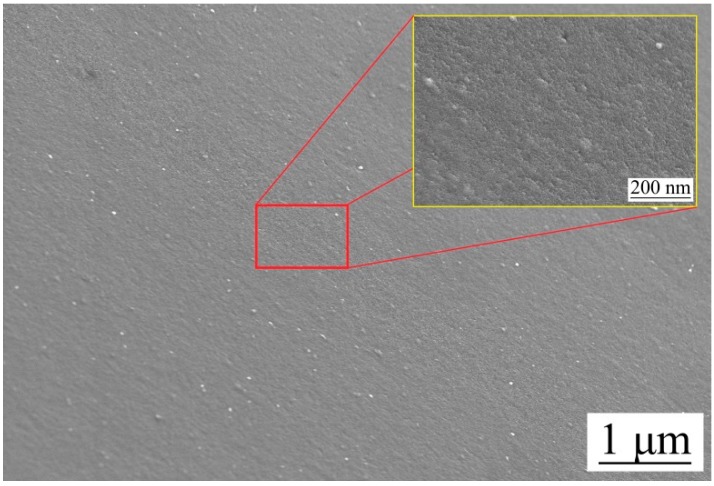
SEM diagram of the original mold after sputtering the Cr layer.

**Figure 12 micromachines-10-00702-f012:**
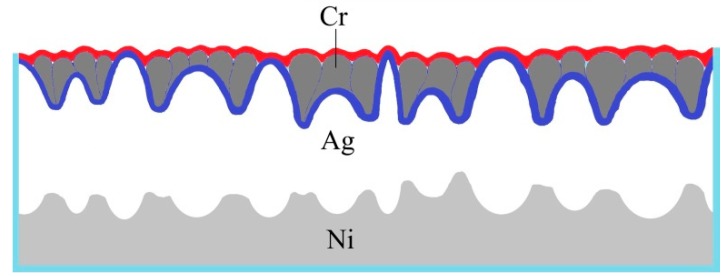
Schematic illustration of “inlay model”.

**Table 1 micromachines-10-00702-t001:** Magnetron sputtering parameters.

Target Material	Power (W)	Time (s)	Temperature (°C)
Cr	200	15	30
Ag	200	300	30

**Table 2 micromachines-10-00702-t002:** Electrodeposition parameters.

Parameters of Electrodeposition	Value
Nickel aminosulfonate	250–450 g·L^−1^
Nickel dichloride	5–30 g·L^−1^
Boric acid	30–40 g·L^−1^
Lauryl sodium sulfate	0.06–0.2 g·L^−1^
current density	4 A·dm^−2^
temperature	50 °C
time	3.5 h
pH	3.5–4.0

**Table 3 micromachines-10-00702-t003:** Testing liquids for evaluating surface energy.

Testing Liquids	Temperature (°C)	γ^p^ (mN/m)	γ^d^ (mN/m)	γ (mN/m)
Pure water	20	51.00	21.80	72.8
Diiodomethane	20	0	50.80	50.8

**Table 4 micromachines-10-00702-t004:** The surface energy of the optical mold before and after being sputtered.

Different States	Contact Angle (°C)		Contact Angle of CB Model (°C)		γ (mN/m)
	Pure Water Diiodomethane		Pure Water Diiodomethane		
Before sputtering	87.56	56.89	88.47	60.25	31.47
After sputtering	97.41	86.73	106.40	86.97	15.53

**Table 5 micromachines-10-00702-t005:** Release strength under different process conditions.

Scheme	Release Strength
1	1.94 N/cm
2	1.43 N/cm

**Table 6 micromachines-10-00702-t006:** Retro-reflection coefficient under different schemes.

Scheme	Retro-Reflection Coefficient
1	1600 cd·lx^−1^·m^−2^
2	2100 cd·lx^−1^·m^−2^

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
