# Peer review of "A Cr Anti-Sticking Layer for Improving Mold Release Quality in Electrochemical Replication of PVC Optical Molds"

_micromachines, 2019, doi:10.3390/mi10100702_

Round 1

Reviewer 1 Report

It will be worthwhile for the authors to explain the benefit of other coating techniques of the Cr layer such as EBeam evaporation, Atomic layer deposition compared to Magnetron Sputtering. The control of the layer uniformity may be better and if this is the case, whether they anticipate a further improvement to the anti-stick properties of the Cr layer.

Author Response

Query:

It will be worthwhile for the authors to explain the benefit of other coating techniques of the Cr layer such as EBeam evaporation, Atomic layer deposition compared to Magnetron Sputtering. The control of the layer uniformity may be better and if this is the case, whether they anticipate a further improvement to the anti-stick properties of the Cr layer.

Response to Query:

It is true that good uniformity of Cr thin films could be obtained by means of atomic layer deposition (ALD) and EBeam evaporation, however these processes would increase the proceeding costs. Magnetron sputtering has the characteristics of simple process and easy control. The thickness of PVC optical mold is only 0.2 mm, and the increase of temperature during the Cr films growth in ALD and EBeam evaporation will lead to its deformation, which seriously affects the optical properties of micro-prisms. The deposition at near room temperature can be realized by magnetron sputtering. Therefore, Magnetron sputtering method is selected to deposit the anti-sticking layer of PVC mold in current study.

Reviewer 2 Report

The problem of demoulding is always of great interest and in recent years the study of techniques that facilitate the extraction in different fields and industrial technologies has increased considerably, especially with the increasing use of micro technologies which produce very small components or features  and that require a fairly accurate extraction and manipulation to prevent damages and breaking.

This article is in continuity with the above arguments and deals with the issue of mold-release from the point of view of the mould coating with nickel and chrome nanoparticles.

The article appears to be well organized as a whole, albeit leaving out in some points some important descriptive aspects that should be improved for greater clarity towards the readers.

First of all, in the introductions it is never nice to see eight citations together as in the case of the sentence on line 32-33. It would be advisable to describe as much as possible or at least mention only those that are really important. I do not believe that eight quotations can be summarized in two lines.

On line 51, you write: "The anti-sticking layer must be thin and uniform, in order to ensure optical properties", however, you do not give me references that attest to this statement, although true.

Then continuing, a series of brief considerations are made on some technologies usually used to deposit the coating such as the SAM and thermal spraying method, deeming them destructive due to the superhydrophobicity of the micro lenses and micro prisms, and then the method used is introduced, magnetron sputtering as a solution to all problems. I believe that this part should be broadened and revised by inserting citations that substantiate the statements and perhaps even inserting some references to the magnetron sputtering technology.

The lines 59-62 would perhaps eliminate them because in the introduction usually information on the conclusions is not already given, this is perhaps the task of the abstract.

In materials and methods I have found the tables related to the parameters of the magnetron sputtering, but I have not noticed then the experimental plan used to study the process whose results are reported in the following paragraphs. Besides these parameters how were they chosen? scientific bibliography, datasheet, etc ...

Furthermore English, in my opinion, needs some revision in some parts.

Below I try to give some further specific suggestions that may help to improve the final quality of the paper:

line 69: I would write: "The anti-sticking layer was obtained by magnetron sputtering, that is schematically explained in Fig.1 ...";

at line 175 when you say "... different deposition schemes ..." what are you referring to? it is not clear to me. A similar issue also applies to line 184;

at line 209 you say: "The average release was needed to separate the nickel mold from the original PVC mold is 208 measured and then the release strength is calculated". How do you measure it? Are you referring to the methods mentioned in chapter 2?

figure 10 shows in the abscissa descriptions when instead it could report "scheme 1" and "scheme 2". to review;

paragraph 4.1, in my opinion, should be reviewed above all in the form. There are many repetitions like "energy", "separation" and "particle". Furthermore, in my opinion, the paragraph running from line 248 to line 255 should also be clarified in reference to figure 11. In fact, from the figure, it is impossible to understand, for example, the reference that is made in the text to the cylindrical surface. In particular, revise the form to lines 253-255;

line 263: "... the original mold is of the atomic level ..." go on "at atomic level.

Author Response

Query 1:

First of all, in the introductions it is never nice to see eight citations together as in the case of the sentence on line 32-33. It would be advisable to describe as much as possible or at least mention only those that are really important. I do not believe that eight quotations can be summarized in two lines.

Response to Query 1:

As suggested by the reviewer, eight quotations are discussed separately on line 29-37.

Query 2:

On line 51, you write: "The anti-sticking layer must be thin and uniform, in order to ensure optical properties", however, you do not give me references that attest to this statement, although true.

Response to Query 2:

References 23, 24 and 25 have been given in the statement of that the anti-sticking layer must be thin and uniform, which are added on line 57.

Query 3:

Then continuing, a series of brief considerations are made on some technologies usually used to deposit the coating such as the SAM and thermal spraying method, deeming them destructive due to the superhydrophobicity of the micro lenses and micro prisms, and then the method used is introduced, magnetron sputtering as a solution to all problems. I believe that this part should be broadened and revised by inserting citations that substantiate the statements and perhaps even inserting some references to the magnetron sputtering technology.

Response to Query 3:

An explanation of EBeam evaporation, thermal spraying method and SAM are added on lines 57-65 in the main text. An explanation of magnetron sputtering technology is broadened as well on lines 65-68.

Query 4:

The lines 59-62 would perhaps eliminate them because in the introduction usually information on the conclusions is not already given, this is perhaps the task of the abstract.

Response to Query 4:

The lines 59-62 are eliminated.

Query 5:

In materials and methods I have found the tables related to the parameters of the magnetron sputtering, but I have not noticed then the experimental plan used to study the process whose results are reported in the following paragraphs. Besides these parameters how were they chosen? scientific bibliography, datasheet, etc ...

Response to Query 5:

An explanation of sputtering process and sputtering parameters, is added in the text as below in section 2.1 (at lines 82-86):

Before sputtering, the chamber was pumped down to 2.5×10-2 Pa, and during sputtering, Ar gas was introduced into the vacuum with vacuum going up to 1.0 Pa. After optimization of the experimental parameters, the sputtering power is 200 W [31,32] and the sputtering temperature is 30 â„ƒ. The sputtering time of the Cr anti-sticking layer is 15 s, and the sputtering time of the Ag conductive layer is 300 s.  

Query 6:

Furthermore English, in my opinion, needs some revision in some parts.

Response to Query 6:

Language has been polished by a senior scientist.

Query 7:

line 69: I would write: "The anti-sticking layer was obtained by magnetron sputtering, that is schematically explained in Fig.1 ...";

Response to Query 7:

As been suggested by the reviewer, the sentence at lines 78-79 is modified as “The anti-sticking layer was obtained by magnetron sputtering, that is schematically explained in Fig.1.”

Query 8:

At line 175 when you say "... different deposition schemes ..." what are you referring to? it is not clear to me. A similar issue also applies to line 184;

Response to Query 8:

At lines 97-100 different deposition schemes are given.

At line 195: Figure 5. Ag surface micromorphology : (a), (b) schemes 1; (c), (d) schemes 2.

Query 9:

At line 209 you say: "The average release was needed to separate the nickel mold from the original PVC mold is 208 measured and then the release strength is calculated". How do you measure it? Are you referring to the methods mentioned in chapter 2?

Response to Query 9:

It is true that the schematic illustration of release test is in the Figure 2. The average value of release force F is recorded by the dynamometer, and the release strength is calculated by =F/L [N/cm] [15,33], at lines 111-112.

Query 10:

Figure 10 shows in the abscissa descriptions when instead it could report "scheme 1" and "scheme 2". to review;

Response to Query 10:

Abscissa descriptions are instead "scheme 1" and "scheme 2" in the Figure 10 (at line 245).

Query 11:

Paragraph 4.1, in my opinion, should be reviewed above all in the form. There are many repetitions like "energy", "separation" and "particle". Furthermore, in my opinion, the paragraph running from line 248 to line 255 should also be clarified in reference to figure 10. In fact, from the figure, it is impossible to understand, for example, the reference that is made in the text to the cylindrical surface. In particular, revise the form to lines 253-255;

Response to Query 11:

The language is modified in the paragraph 4.1.

Lines from 248 to 255 are clarified in reference to figure 10 (at line 244).

At lines 256-257, reference 42 and 43 are given.

Query 12:

line 263: "... the original mold is of the atomic level ..." go on "at atomic level.

Response to Query 12:

The sentence at line 268 is modified to:

“... the original mold is at atomic level ...”

Reviewer 3 Report

The manuscript reports the effects of 3-nm-thick Cr thin film for mold-release quality in electrochemical replication of PVC molds. The Cr layer acted as anti-sticking layer and resulted in the lower release strength of electrochemically prepared Ni molds. In addition, the insertion of the Cr thin film resulted in the smoother surface of released micro-prisms array Ni molds led to the higher retro-reflection. The results are interesting and can be applied for practical fabrication process of optical micro arrays using PVC molds. However, I found several points should be addressed before publication. The manuscript should be reconsidered for publication after major revisions. Hereafter, I listed my specific comments.

Why the Cr was selected as the anti-sticking layer? Regarding to the fabrication process, were the molds exposed to air atmosphere after the deposition of Cr thin films prior to the deposition of Ag? or the deposition of Cr and Ag had been continued without taking out from the sputtering chamber? If the former case is, the surface of Cr thin film may be oxidized or hydroxylated. Does such termination at early stage of metal deposition cause some effects for the process? It should be clarified whether a deposition of very thin Ag film on the PVC followed by a main deposition of Ag cause some effects or not. Since the mold release process seems happened at the interface between Ag and PVC in Scheme1 and between Cr and PVC in Scheme2, respectively. The differences between deposition of Cr on PVC and Ag on PVC should be discussed more in details. What is error (or distribution) of the surface roughness measured by the laser confocal microscope. In addition, please clarify how long distance was scanned by the microscope for the roughness measurements. The Ra roughness of the original PVC mold was about 600 nm (0.60 μm). The Ra roughness decreased to 550 nm (0.55 μm) by the deposition of 3-nm-thick Cr thin film by the magnetron sputtering. It was explained that the sunken holes were partially filled by deposited Cr. However, it is difficult to understand that such the large Ra value in the level of several hundred nm are influenced by the film deposition with thickness of only 3 nm. For instance, the scale of surface profile in Fig. 4 (a) is ~ 10,000 nm (~ 10μm). I cannot imagine the deposition of film with a thickness of 3 nm influence the profiles in μmscale. It should be more clearly explained why the surface roughness decreased by the deposition of extremely thin Cr layer. If I understand correctly, the surface morphology of Fig. 8 (b) is the released side of Ni molds, that is, the released Cr surface. The surface by Scheme2 was obviously smoother than that of Scheme1. But the Ra values are lacking for these layers. Is Fig. 3 (a) the SEM image measured without any surface treatment? I just worry about charging effects. At line 73 and 74, the unit of Pa should be used instead of hPa. At line 198, it is mentioned that “The two surface ….. are measured by energy spectrum …”. What is the energy spectrum? Please clearly indicate the experimental method. It would be better to use retro-reflection or retroreflection instead of reflection for clarity. The reflection coefficient in Table 6 should be retro-reflection coefficient as well. At line 165 and 169 “PVC model” would be “PVC mold”.

Author Response

Query 1:

Why the Cr was selected as the anti-sticking layer?

Response to Query 1:

There are several reasons to select Cr as the anti-sticking layer. First, Cr is an inactive metal, which is stable for oxygen and moisture at room temperature; second, thin and uniform Cr films can be obtained by physical deposition; third, the surface energy of Cr films is lower than that of PVC material, and the releasing of the nickel mold from the PVC mold becomes easier resulting in better mold-release quality. In the literature “Electroforming for Replicating Nanometer-Level Smooth Surface, Journal of Nanoscience and Nanotechnology, Vol. 11, 2886-2889, 2011”, Cr atoms are deposited by an arc plasma deposition method and the Cr films act as a binding material and smooth surface.

Query 2:

Regarding to the fabrication process, were the molds exposed to air atmosphere after the deposition of Cr thin films prior to the deposition of Ag? or the deposition of Cr and Ag had been continued without taking out from the sputtering chamber? If the former case is, the surface of Cr thin film may be oxidized or hydroxylated.

Does such termination at early stage of metal deposition cause some effects for the process?

Response to Query 2:

Regarding to the fabrication process, the molds is exposed to air atmosphere after the deposition of Cr thin films and prior to the deposition of Ag. It is true that the surface of Cr thin film may be oxidized or hydroxylated. However, during the transport, the samples were protected with nitrogen atmosphere. Moreover, the obvious oxidation regions were not found by SEM observation, and the surface quality meets the needs of industrial requirements.

Query 3:

Since the mold release process seems happened at the interface between Ag and PVC in Scheme1 and between Cr and PVC in Scheme2, respectively. The differences between deposition of Cr on PVC and Ag on PVC should be discussed more in details.

Response to Query 3:

It is true that the differences between deposition of Cr on PVC and Ag on PVC should be discussed in more details. A simple explanation of deposition of Cr on PVC and Ag on PVC, as below is added in section 2.1 (at lines 94-97):

This work mainly studies the releasing effect under different schemes. The process parameters and thickness of deposition of Cr on PVC and Ag on PVC are different. The Cr layer is relatively thin, which can reduce the surface energy and help releasing. The Ag layer is thicker, and its role is to act as the conductive layer of electroforming Ni. 

Query 4:

What is error (or distribution) of the surface roughness measured by the laser confocal microscope. In addition, please clarify how long distance was scanned by the microscope for the roughness measurements. The Ra roughness of the original PVC mold was about 600 nm (0.60 μm).

Response to Query 4:

Ra is the arithmetic mean deviation of the measured surface contour. Six measurements per test position and average of six values are given. Among them, three times were measured along the bottom edge of the micro-prisms, and the sampling lengths were 120 μm, 80 μm and 60 μm, respectively. Three times were measured vertical the bottom edge of the micro-prisms, and the sampling length is 100 μm, 80 μm and 60 μm, respectively.

Query 5:

The Ra roughness decreased to 550 nm (0.55 μm) by the deposition of 3-nm-thick Cr thin film by the magnetron sputtering. It was explained that the sunken holes were partially filled by deposited Cr. However, it is difficult to understand that such the large Ra value in the level of several hundred nm are influenced by the film deposition with thickness of only 3 nm. For instance, the scale of surface profile in Fig. 4 (a) is ~ 10,000 nm (~ 10μm). I cannot imagine the deposition of film with a thickness of 3 nm influence the profiles in μm scale. It should be more clearly explained why the surface roughness decreased by the deposition of extremely thin Cr layer.

Response to Query 5:

The thickness of Ag layers is 500 nm (at line 86), and the average particle size of Ag particles is 230 nm, that majorly effects the surface roughness. Cr layer mainly plays an anti-adhesion role, that is, to prevent the occurrence of microtearing during releasing. At the same time, Cr deposited particles will fill the gap of Ag particles, which smoothen the surface of the Ni mold at the micro level. The functions of Cr anti-adhesive layer are explained by the adhesion energy and surface morphology in this study.

Query 6:

If I understand correctly, the surface morphology of Fig. 8 (b) is the released side of Ni molds, that is, the released Cr surface. The surface by Scheme2 was obviously smoother than that of Scheme1. But the Ra values are lacking for these layers. Is Fig. 3 (a) the SEM image measured without any surface treatment?

Response to Query 6:

The original mold surface is the surface before Ni deposition shown in Figure 3, and for both schemes, they are the Ag surface.

Query 7:

I just worry about charging effects. At line 73 and 74, the unit of Pa should be used instead of hPa. At line 198, it is mentioned that “The two surface ….. are measured by energy spectrum …”. What is the energy spectrum?

Response to Query 7:

At line 82 and 83, The unit is modified to “Pa”.

At line 212, “energy spectrum” is modified to “Energy Dispersive Spectrometer”.

Query 8:

Please clearly indicate the experimental method. It would be better to use retro-reflection or retroreflection instead of reflection for clarity. The reflection coefficient in Table 6 should be retro-reflection coefficient as well. At line 165 and 169 “PVC model” would be “PVC mold”.

Response to Query 8:

In Table 6, at lines 23, 163-165, 229-233, 241, and in Fig.10, “reflection” is modified to “retro-reflection”.

At line 179 and 183 “PVC model” is modified to “PVC mold”.

Round 2

Reviewer 2 Report

The authors accepted and improved all the suggestions reported in the previews review form. Now the paper is clearer and well supported by bibliography and explanations. 

In my opinion now can be accepted in the present form. I just suggest the authors to add, in the final part of the introduction, a summary or a rearrangement of what it has been reported at lines 91-94, where the research topic is well-motivated. 

Reviewer 3 Report

The issues pointed out by reviewers have been addressed. I think the manuscript has been significantly improved by the revision. I recommend publication at the present form in the journal.